# Transmembrane Protein LMxysn_*1693* of Serovar 4h *Listeria monocytogenes* Is Associated with Bile Salt Resistance and Intestinal Colonization

**DOI:** 10.3390/microorganisms10071263

**Published:** 2022-06-21

**Authors:** Fanxin Jin, Youwei Feng, Chao Chen, Hao Yao, Renling Zhang, Qin Zhang, Fanzeng Meng, Xiang Chen, Xin’an Jiao, Yuelan Yin

**Affiliations:** 1Jiangsu Key Laboratory of Zoonosis, Yangzhou University, 48 East Wenhui Road, Yangzhou 225009, China; 18762314653@163.com (F.J.); 18705275306@163.com (Y.F.); m19551656353@163.com (C.C.); YH1163636424@163.com (H.Y.); renlingz9313@gmail.com (R.Z.); 17748044125@163.com (Q.Z.); menfanzeng@icloud.com (F.M.); chenxiang@yzu.edu.cn (X.C.); jiao@yzu.edu.cn (X.J.); 2Key Laboratory of Prevention and Control of Biological Hazard Factors (Animal Origin) for Agrifood Safety and Quality, MOA of China, Yangzhou University, 48 East Wenhui Road, Yangzhou 225009, China; 3Joint International Research Laboratory of Agriculture and Agri-Product Safety, Yangzhou University, 48 East Wenhui Road, Yangzhou 225009, China; 4Jiangsu Co-Innovation Center for Prevention and Control of Important Animal Infectious Disease and Zoonosis, Yangzhou University, 48 East Wenhui Road, Yangzhou 225009, China

**Keywords:** *Listeria monocytogenes*, bile salt, virulence, biofilm, ABC transporter, genomic island

## Abstract

*Listeria monocytogenes* (Lm) is a ubiquitous foodborne pathogen comprising of 14 serotypes, of which serovar 4h isolates belonging to hybrid sub-lineage Ⅱ exhibit hypervirulent features. LMxysn_*1693* of serovar 4h Lm XYSN, a member of genomic island-7 (GI-7), is predicted to a membrane protein with unknown function, which is conserved in serovar 4h *Listeria monocytogenes*. Under bile salts stress, Lm XYSN strain lacking LMxysn_*1693* (XYSN∆*1693*) exhibited a stationary phase growth defect as well as a reduction in biofilm formation and strikingly down-regulated bile-salts-resistant genes and virulent genes. Particularly, LMxysn_1693 protein plays a crucial role in Lm XYSN adhesion and invasion to intestinal epithelial cells, as well as colonization in the ileum of mice. Taken together, these findings indicate that the LMxysn_*1693* gene encodes a component of the putative ABC transporter system, synthetically interacts with genes involved in bile resistance, biofilm formation and virulence, and thus contributes to *Listeria monocytogenes* survival within and outside the host.

## 1. Introduction

*Listeria monocytogenes* (Lm) is ubiquitously distributed in diverse environments, which can survive in extreme conditions at a low temperature −0.4 °C–45 °C [1], wide pH range (4.5–9.0) and high concentration of 10% NaCl [2]. Lm is a facultative intracellular pathogen that is responsible for gastroenteritis, meningitis and maternofetal infections [3], and the mortality rate reaches up to 20% to 30% [4]. It is presently assumed that foodborne outbreaks are increasingly caused by hypervirulent complex clones (CCs) such as CC1, CC2, CC4, and CC6 with high clinical frequency [5]. CC33, consisting of hybrid sub-lineage Ⅱ (HSL-Ⅱ) hypervirulent isolates, is an emerging complex clone with 200–400 folds higher virulence than Lm EGD-e via oral infection [6], which is highly susceptible to wild-type mice. While early work suggested that 10^8^–10^10^ CFU were normally needed to establish intestinal infection in mice, it was shown that the oral infectious dose of CC33 isolates was 10^5^ CFU, which suggested that their unique virulence genes were involved in breaking through the intestinal barrier and responsible for making mice a susceptible host.

Multiple *Listeria* genes associated with resisting heat (*dnaK*, *groES*, *clpC*, *clpP* and *clpE*), cold (*cspA*, *cspB* and *cspD*) and acid (*sigB*) contribute to its survival in harsh environmental conditions [7]. Once Lm enters the gastrointestinal tract, bacteria cells face several threats, such as trypsin, stomach acid, bile salts and inflammatory responses. Moreover, some genes are related to the resistance of acidic environment in Lm, including the glutamate decarboxylase system (GAD) [8], the F0F1-ATPase system [9] and Sigma B (SigB) [10]. Additionally, the *bsh* gene, regulated by *prfA* and *sigB*, encodes the bile salt hydrolase (BSH), which is essential for bacteria to break through the gastrointestinal barrier. BSH can hydrolyze bile salts into amino acids and bile acids to protect bacteria [11]. Furthermore, BilE and bile regulatory transcription factor A (BrtA) could also enhance bile salts tolerance and colonization in the liver and gallbladder [12]. However, how hypervirulent strains resist the stress and survive in the gastrointestinal tract remains unknown.

Bacteria could obtain new phenotypic characteristics through genomic islands, which are of great significance for pathogenicity and antibiotic resistance, and even form new bacterial pathogenic subspecies. Genome islands are often closely related to horizontal gene transfer and are the basis and source for discovering and identifying novel functional islands such as new pathogenicity islands, resistance islands, metabolic islands and symbiotic islands. Kovačević et al., found that a 50 kb genomic island LGI1 carrying resistance gene to cefoxitin and nalidixic acid in three strains of 1/2a *Listeria* [13]. Lee et al., identified that a 4b clonal complex could resist arsenic and cadmium with a 35 kb genomic island. Moreover, metabolic islands have also been detected in other pathogenic bacteria [14]. For example, Salmonella Senftenberg could survive under glucose as the sole carbon source, mainly due to the metabolic island CTnscr94 with related genes, which is crucial for enhancing environmental adaptability [15]. Undoubtedly, CC33 isolate Lm XYSN harbor eight genomic islands essentially involved in the adaption of saprophytic and parasitic life [6]. Understanding functions of genes in the genomic islands contribute to deciphering the pathogenic mechanism from virulence to hypervirulence.

LMxysn_*1693* is a member of genomic island-7 (GI-7) carrying 20 ORFs (LMxysn_*1677*-LMxysn_*1696*), and most genes of GI-7 are hypothetical proteins with unknown function. In this study, we found that LMxysn_*1693* was involved in up-regulating bile resistant and virulence genes, thereby facilitating resistance to bile salts and enhancing colonization in mice. Our work paves the way to understand the role of genomic island and the hypervirulence of CC33 isolates.

## 2. Materials and Methods

### 2.1. Bacterial Strains, Cell Lines and Animals

High-virulent Lm XYSN was isolated from a case of ovine listeriosis [6]. Shuttle vector pAULA was kindly donated by Prof. Chakraborty (Justurs Liebig University, Giessen, Germany). All Lm strains were cultured in brain heart infusion (BHI; Becton Dickinson, Sparks, MD, USA). The Caco-2 BBe cell line was propagated in the Dulbecco’s modified Eagle medium (DMEM; Gibco, Waltham, MA, USA) supplemented with 10% fetal bovine serum (FBS; Gibco, Waltham, MA, USA). Six-week-old female C57BL/6 mice were purchased from Vital River Laboratory Animal Technology Co., Ltd. (Beijing, China). Animal experiments were conducted by following guidelines laid down for the welfare and ethics of experimental animals. All animals were kept at the animal biosafety facilities according to procedures approved by the Institutional Animal Ethics Committee of Yangzhou University (reference number 202205007).

### 2.2. Bioinformatics Analysis

IslandViewer4 (creator Brinkman lab, Burnaby, BC, Canada) was used to predictive analysis for genomic islands (https://www.pathogenomics.sfu.ca/islandviewer/resources/ (accessed on 13 November 2021)). Nucleobase sequence and amino acids of LMxysn_1693 were compared with database to search for homology gene and proteins via Blast (creator National Institutes of Health, Bethesda, MD, USA) (https://blast.ncbi.nlm.nih.gov/Blast.cgi (accessed on 15 November 2021)). The domains of proteins of GI-7 were predicted through SMART (version 9, Heidelberg, Germany) (http://smart.embl-heidelberg.de, accessed on 20 November 2021) and InterPro (version 87.0, creator European Bioinformatics Institute, Cambridge, Britain) (https://www.ebi.ac.uk/interpro/ (accessed on 20 November 2021)). Signal peptide and transmembrane structure of all proteins were analyzed using SingnalP-6.0 (https://services.healthtech.dtu.dk/service.php?SignalP-6.0 (version 6.0, creator Technical University of Denmark, Lyngby, Denmark) (accessed on 20 November 2021)) and TMHMM-2.0 (https://services.healthtech.dtu.dk/service.php?TMHMM-2.0 (version 2.0, creator Technical University of Denmark, Lyngby, Denmark) (accessed on 20 November 2021)), respectively. Amino acid sequence of LMxysn_1693 protein was submitted to the SWISS-MODEL (creator SIB Swiss Institute of Bioinformatics and the Biozentrum of the University, Basel, Switzerland) to obtain the hypothetical 3D structure (http://swissmodel.expasy.org (accessed on 22 November 2021)).

### 2.3. Mutant and Complemented Strains Construction

Construction of mutant and complemented strains was performed as previously described [16]. Briefly, LMxysn_*1693* gene flanking regions for achieving homologous recombination were amplified and ligated with digested pAULA by clone express Ⅱ one-step cloning kit (Vazyme, Nanjing, China). The recombinant plasmid pAULA-*1693*-SX was identified and transferred into competent strain Lm XYSN by electroporation to generate the mutant strain XYSNΔ*1693*. The complemented strain XYSNΔ*1693*::*1693* was obtained on the basis of the deletion strain according to the same method within plasmid pAULA-*1693*H-SX, of which synonymous mutations at two amino acids following the termination codon of the LMxysn_*1693* gene. All the primers used for constructing mutant and complemented strains in this study are listed in Appendix A.

### 2.4. Evaluation of Biological Characteristics

Lm XYSN, XYSNΔ*1693* and XYSNΔ*1693*::*1693* were cultured on BHI plate for 24 h. The next day, the bacteria cells were scraped from the plate using an inoculating loop and transferred into 5 mL of 0.45% normal saline. The bacterial turbidity was controlled at ∼0.5 with a nephelometer and biochemical characteristics were identified using a VITEK^®^2 GN ID card (Biomerieux, Marcy-l’Étoile, France).

### 2.5. Growth Curve Analysis

Bacterial of exponentially growing cultures of Lm XYSN, XYSNΔ*1693* and XYSNΔ*1693*::*1693* were harvested and adjusted to an initial OD_600_ value of 0.05 in BHI culture and 0.2% bile salt (Merck Sigma, Darmstadt, Germany) in BHI media, respectively. Three parallel replicates were set for each strain and each treatment with 20 mL media for 12 h. The OD_600_ value of each flask was measured at an hour interval for BHI culture and 0.2% bile salt in BHI culture treatment.

### 2.6. Determination of Biofilm Formation

The ability of bacterial biofilm formation was conducted by crystal violet assay. The overnight cultures of XYSN, XYSNΔ*1693* and XYSNΔ*1693*::*1693* were washed twice, then the OD_600_ was adjusted to 1.0 and diluted 10 times with BHI medium or bile salt medium. Following this, 200 μL bacteria culture was added to the wells of a 96-well plate and incubated at 4 °C, 37 °C or 42 °C for 48 h, respectively. The medium was discarded and stained with 0.1% crystal violet for 15 min, then crystal violet was removed and washed three times with ultrapure water. After drying at 56 °C, 100 μL 96% ethanol was added to each well for 15 min for elution. The biofilm formation was measured at 595 nm by BioTek synergy 2 enzyme-labeled instrument (BioTek, Winooski, VT, USA).

### 2.7. Analysis of Gene Expression at Transcriptional Level

Total RNA from exponentially growing cultures was extracted with RNAprep pure Cell/Bacteria Kit (Tiangen, Beijing, China). Then, 1 μg RNA was reverse transcribed into cDNA through the PrimeScript RT reagent kit (Takara, Beijing, China). The quantitative real-time PCR (qRT-PCR) operation program followed the recommended thermal cycling conditions of 7500 Real Time PCR System (Applied Biosystems, Waltham, MA, USA) under the following cycling conditions: 95 °C for 10 min, 40 cycles (95 °C for 15 s, 60 °C for 1 min) with AceQ Universal SYBR qPCR Master Mix (Vazyme, Nanjing, China). The house-keeping gene *gyrB* was selected as the internal reference gene to reflect the differences in gene expression objectively. The 2^ΔΔCt^ (ΔCt = Ct*_objective gene_* − Ct*_reference gene_*) method was used to calculate the relative changes. All the primers used for qRT-PCR in this study are listed in Appendix A.

### 2.8. Adhesion, Invasion and Replication Capacity of Lm

Human intestinal epithelial cells Caco-2 BBe with stabilized growth conditions were seeded into 24-well cell culture plates. Freshly cultured cells of Lm XYSN, XYSNΔ*1693* and XYSNΔ*1693*::*1693* were added at a bacterium/cell ratio (multiplicity of infection, MOI) of 20. After incubating for 1 h at 37 °C with 5% CO_2_, the medium was removed, and the cells were lysed in 0.2% Triton X-100 buffer for 9 min to release the bacteria. The number of bacteria was assessed after serial dilutions of the lysates onto agar plates to calculate the adhesion ability. Additionally, the cells were incubated for another 15 min and 2 h, respectively, with DMEM medium (containing 50 μg/mL gentamicin sulfate) to determine the invasion and proliferation.

### 2.9. Infection to Mice

Six-week-old C57BL/6 female mice (n = 5/group) were starved for 12–16 h before treatment (water allowed) and orogastrically inoculated with Lm XYSN, XYSNΔ*1693* and XYSNΔ*1693*::*1693*. Briefly, the mice were starved for 12 h before treatment (water was allowed) and challenged with approximately 3 × 10^6^ CFU of bacteria at a volume of 500 μL (containing 30 mg/mL CaCO_3_) per mouse. Bacterial loads in the spleen, liver, colon and ileum were evaluated at 72 h post-infection (p.i.). Intestine samples were treated as follows: about 2-cm-length intestine was taken and washed by a syringe with 5 mL of phosphate-buffered saline (PBS) to remove the luminal contents. The organs were homogenized and plated serial dilutions on the BHI plate, the intestine samples were spread onto modified Oxford medium (Becton Dickinson, Sparks, MD, USA) chromogenic plate, then the bacterial loads were counted after 20 h post spreading.

### 2.10. Date Analysis

Statistical analyses were performed with GraphPad Prism 8 (GraphPad Software, version 8.0.1, San Diego, CA, USA). The results were expressed as the means ± SD for the results of all qRT-PCR. The results of detecting adhesion, invasion and replication capacity of Lm were expressed as such as well. Values were means ± SEM for the results of growth curves and *in vivo* experiments. Data were analyzed by Student’s *t*-test or Tukey’s multiple comparisons test. Differences were considered significant at ns (no significance), * *p* ≤ 0.05, ** *p* ≤ 0.01, *** *p* ≤ 0.001 and **** *p* < 0.0001.

## 3. Results

### 3.1. Association with the Transporter System

LMxysn_*1693*, a previously uncharacterized 534 bp-length gene located on GI-7 harboring 20 genes (Figure 1A) was a conservative and unique gene in CC33 (HSL-II) *Listeria* strains. The bioinformatics analysis predicted that LMxysn_*1693* encoded a transmembrane protein (Appendix A) could form a unique shape dimer (Figure 1C). Interestingly, the adjacent genes LMxysn_*1691* and LMxysn_*1692* were predicted to encode multiple spanning transmembrane barrel proteins to form hydrophobic channels (Table 1). Moreover, LMxysn_*1691* was predicted to be a membrane electron transporter, and LMxysn_*1692* encoded a multisubunit Na^+^/H^+^ antiporter (Table 1). Additionally, LMxsyn_*1695* encoded a N-acetylmuramoyl-L-alanine amidase. These data supported that LMxysn_1691, LMxysn_1692 and LMxysn_1693 were possible to form an ABC transporter system, GI-7 possibly composed of two ABC transporters, which were involved in transporting extracellular and cell surface proteins (Figure 1B and Appendix A).

To evaluate the function of LMxysn_*1693*, we constructed XYSNΔ*1693* and XYSNΔ*1693*::*1693* (Appendix A) and detected the transcriptional expression level of the genes in GI-7. Interestingly, LMxysn_*1693* could up-regulate the expression of almost all the genes, such as LMxysn_*1691*, LMxysn_*1694* and LMxysn_*1695* (Figure 1D and Appendix A). Our results suggested that LMxysn_*1693* could regulate multiple genes in the cluster, thus playing a crucial role in the function of GI-7.

To evaluate the role of LMxysn_1693 protein in cell homeostasis, the structures of XYSN, XYSNΔ*1693* and XYSNΔ*1693*::*1693* were observed by transmission electron microscopy. We found that there was no obvious ultrastructural change in the cell wall surface among the three strains (Appendix A). Additionally, the mutant exhibited similar sensitivity to penicillin, chloramphenicol, polymyxin B and kanamycin (Appendix A). In summary, these data suggested that LMxysn_*1693* was possibly not involved in *Listeria* cell remodel. The 43 biochemical characteristics of Lm conducted by the automatic microbial identification instrument verified that LMxysn_*1693* might not be involved in metabolism of XYSN (Appendix A).

### 3.2. Involvement in Resisting Bile Salt

The growth curves of XYSN, XYSNΔ*1693* and XYSNΔ*1693*::*1693* (Figure 2A) were generally similar when culturing in BHI at 37 °C. While the growth of the mutant was significantly inhibited in the stationary phase cultured in the BHI supplemented with 0.2% bile salt (Figure 2B). The treatment of bile salts affected the growth of the mutant in the stationary phase, which indicated that LMxysn_*1693* contributed to resisting the stress response.

### 3.3. Contribution to Biofilm Formation

Biofilms are composed of polysaccharides, lipoproteins, fibrin and other component secreted from the cell, which assist bacteria attach to the surface to resist various adverse factors in the environment, such as high temperature, acid and poor nutrition. Here, we assumed that LMxysn_*1693* was associated with biofilm formation at 4 °C, 37 °C and 42 °C in BHI medium and bile salt medium. We found that the biofilm strength of the XYSN mutant was reduced dramatically compared with the WT and complete strains at 42°C (*p* < 0.05), but there was no difference at 4 °C or 37 °C (Figure 3A). Unexpectedly, in the bile salt medium, the biofilm formation of XYSNΔ*1693* decreased distinctly under the above three culturing temperatures (Figure 3B). These results indicated that deletion of LMxysn_*1693* formed less biofilm in bile salt conditions, therefore decreasing resistance to bile salts stress.

### 3.4. Interaction with Virulence and Bile-Salt-Resistant Genes

Generally, *bsh* encoded bile salt hydrolase, while *brtA* and *sigB* were involved in regulating coupled genes related to bile salt resistance. All the three genes were used to evaluate the interaction between LMxysn_*1693* and bile salt resistance genes. We compared the transcriptional levels of these three genes within mutant, wild-type and reversion when they were cultured in BHI or BHI supplemented with 0.2% bile salt. Interestingly, *bsh* was remarkably down-regulated in the mutant under two culturing conditions (*p* < 0.01) (Figure 4A,B), whereas *brtA* and *sigB* had no significant changes between the mutant and the parental strain. The qRT-PCR results indicated that LMxysn_*1693* could up-regulate the expression of *bsh*, thus enhancing the survival ability in the bile salt environment.

Furthermore, we detected the potential interaction between LMxysn_*1693* and virulent genes *prfA*, *actA*, *ami* and *hly*. *prfA*, as the encoding virulent regulator, could not only play key roles in *Listeria* infection and pathogenicity but also contribute to survival in bile and bile salt conditions. Notably, the expression of *prfA* in the mutant has been down-regulated in both media, even significantly inhibited in the 0.2% bile salt-BHI (Figure 4C). Furthermore, *ami*, *actA* and *hly* were obviously down-regulated in both culturing conditions (Figure 4D). These results suggested that LMxysn_*1693* could regulate both bile salt resisting genes and virulence genes.

### 3.5. Enhancement Adhesion and Invasion Capacity

The foodborne pathogen Lm is able to cause systemic infections by crossing the intestinal barrier colonize intestinal tracts and subsequently crosses the intestinal barrier. The Caco-2 BBE cell is derived from human colon cancer epithelial cells, and its structure and function are similar to those of human small intestinal, so it has been recognized as a model for studying bacterial infection *in vitro*. In this study, the Caco-2 BBE cell was selected to evaluate the infection ability of *Listeria* strains. The mutant significantly reduced the adhesion, invasion and proliferation abilities compared with XYSN, XYSNΔ*1693* and XYSNΔ*1693*::*1693* (*p* < 0.05, *p* < 0.001, *p* < 0.05) (Figure 5A–C). These results strongly supported that the LMxysn_*1693* played crucial role in XYSN infection epithelial cells.

### 3.6. Promotion of Listeria Colonization

Mice are not naturally susceptible to Lm, whereas a low dose of hypervirulent XYSN can cause listeriosis via oral infection. We selected C57BL/6 mice for oral infection to assess the role of gene colonization *in vivo*. After 72 h p.i., the spleen, liver, colon and ileum of the mice were homogenized, and CFUs were counted, for which the results were analyzed through the Tukey’s multiple comparisons test. Just as expected, the bacterial load of the LMxysn_*1693*-deficient strain in the ileum was 100-fold lower than that of the wild-type strain (Figure 6), indicating that the loss of the gene impaired Listerial invasion and colonization ability to small intestine.

## 4. Discussion

The gastrointestinal tract contains a variety of stress factors, including proteolytic enzymes, gastric acid, high osmotic pressure, cytokines and bile salts, etc. [17]. Lm is a foodborne pathogen that has evolved a variety of strategies to survive in the gastrointestinal environment. The secretion of extracellular polysaccharides to form a biofilm is one of the ways to resist the stress factors in the gut environment [18]. Previous studies have demonstrated that transcription activators PrfA, SigB, DegU, the quorum sensing (QS) system and the accessory gene regulatory system (Agr) operon could up-regulate biofilm formation, and down-regulate by ABC transfusion permease and LuxS [19]. Notably, in this study, we find that LMxysn_*1693* can up-regulate the transcriptional expression of PrfA, which indicates that the interaction between LMxysn_1693 and PrfA contributes to an increase in the biofilm formation ability. Interestingly, under bile salt stress, the biofilm formation ability of Lm is strikingly increased, and LMxysn_*1693* further enhances the producing ability, which suggests that biofilm formation is associated with effector genes responding to environmental changes [20]. In brief, our study demonstrates that the LMxysn_*1693* gene is involved in biofilm formation, thus contributing to Lm resistance in the intestinal environment.

Lm can enter and replicate in the lumen of the gallbladder of infected animals, whereas relatively little is known about the mechanisms of survive and growth in this location [21]. Bile salts are the major components in bile to degrade lipid-containing membranes, which have been synthesized by cholesterol in the liver and secreted from the gallbladder into the upper small intestine [11,22]. Thereby, confronting the bile-rich environments is critical for Lm persistence and survival in the gastrointestinal tract. Until now, multiple reported genes (*bsh*, *cad*C, *mdrM*, *mdrT*, *brtA,* etc.) have been elucidated to play crucial roles in enhancing bile salt tolerance and the colonization of Lm [11,23,24,25]. Among these, the bile exclusion locus *bilE,* coordinately regulated by SigB and principal virulence regulator PrfA, plays an essential role in intestinal colonization and virulence [26]. Bile sensor BrtA controls the expression of the cholic acid efflux pump MdrT [27], and CadC represses BSH expression to avoid the over expression of the cholic acid efflux pump MdrT [12]. Meanwhile, the mechanism to resist bile or bile salts for Lm is not fully understood. In this study, we find that LMxysn_*1693* is involved in resistance to bile salt stress through up-regulation with *prfA* and *bsh*. We deduce that LMxysn_*1693* possibly promotes the degradation or efflux pumping of bile salts in *Listeria* cells through activating a regulator of PrfA and unknown proteins, thereby impairing bile toxicity toward bacteria.

PrfA and SigB are global regulators in Lm, which are associated with biofilm formation and bile salt resistance, as well as coordinating with a variety of virulence factors, i.e., genes involved in adhesion, invasion, replication and cell-to-cell translocation [28,29,30]. The present study reveals that LMxysn_*1693* can participate in regulating the expression of *prfA*, key virulence factor *hly* and autolysin-adhesin *ami*, thus ultimately affecting the expression of various virulence factors. Our work verifies that LMxysn_*1693*, as a unique gene distribution in all CC33 *Listeria* strains, not only participates in biofilm formation and bile salt resistance, but also involves adherence and invasion in to intestinal epithelial cells, therefore crossing the intestine barrier and colonizing the small intestines of mice. Based on these data, we propose that LMxysn_*1693* plays an important role for CC33 strains to break through the intestinal barrier, ultimately improving the ability of survival and colonization in the gastrointestinal tract [11,20,24,25,26].

ABC transporters play multiple functions in bacterial metal utilization [31], biofilm formation [32,33,34], resistance to various stress [35,36,37,38,39], virulence and infection [40,41]. Diverse ABC transport systems have been reported in Lm, which are composed of four core members: ATP-binding proteins containing NBDs domains work as dimers, and two transmembrane TMD domain-containing proteins with a short cytoplasmic helix can transport substrate across the cell membrane, among which, NBDs and TMDs can be formed by either homodimers or heterodimers [42]. The energy for the ABC transporters is from hydrolyzing ATP, electron translocation or osmotic pressure. Moreover, extracellular substrate binding protein (SBP) is also required for capturing and delivering substrates to ABC transporters. Interestingly, LMxysn_1691 protein and LMxysn_1692 protein own TMDs with alpha helices, in which the LMxysn_1692 protein features a short cytoplasmic helix. Moreover, LMxysn_1693 is a transmembrane protein forming a uniquely shaped dimer. In our study, based on structural and function predication, we deduce that LMxysn_*1693* is a member of the potential ABC transporter involved in delivering the substrates to anchoring on the cell surface or outside of the cells, such as biofilm formation components, virulent proteins (ActA) and bile salt hydrolase (BSH). Notably, we find that LMxysn_*1693* interacts with multiple genes within and outside of GI-7, plays an important role in biofilm formation, bile salt resistance and virulence, enhancing the survival and colonization capacity of Lm in a gastrointestinal environment.

Taken together, our study suggests that membrane protein LMxysn_1693 is possibly the member of one transporter complex, which may be involved in delivering extracellular proteins. LMxysn_*1693* not only interacts with most genes in GI-7 but also up-regulates the expression of bile-resistance genes and virulence genes, which plays important roles in biofilm formation, bile salt resistance and virulence, therefore contributing to Lm invasion and colonization in the intestine of mice. In a word, LMxysn_*1693* plays a critical role in hypervirulent CC33 isolates survival in the gut, infection and pathogenicity. Further research will be necessary to decipher the coordination mechanism and role of LMxysn_*1693* in enhancement Lm survival within and outside the host.

## Figures and Tables

**Figure 1 microorganisms-10-01263-f001:**
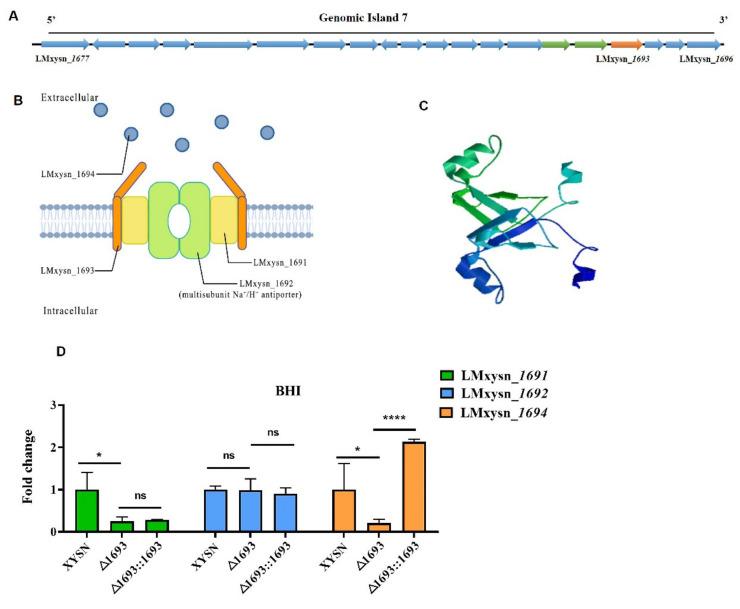
**Bioinformatics analysis of XYSN GI-7.** All genes located in GI-7 are shown in (**A**). The cellular localization and function of the proteins encoded by LMxysn_*1691*, LMxysn_*1692* and LMxysn_*1693* were predicated via Biorender (Toronto, Canada) (https://biorender.com/ (accessed on 5 May 2022)) (**B**). Three-dimensional structure was predicted of the protein encoded by LMxysn_*1693* gene (**C**). The expression levels of Lmxysn_*1691*, Lmxysn_*1692* and Lmxysn_*1694* genes were detected with LMxysn_*1693* deletion strain by qRT-PCR (**D**). Error bars represent SD, n = 3 independent experiments. Statistical analysis was performed using Student’s *t*-test: ns, no significance; * *p* < 0.05, **** *p* < 0.0001.

**Figure 2 microorganisms-10-01263-f002:**
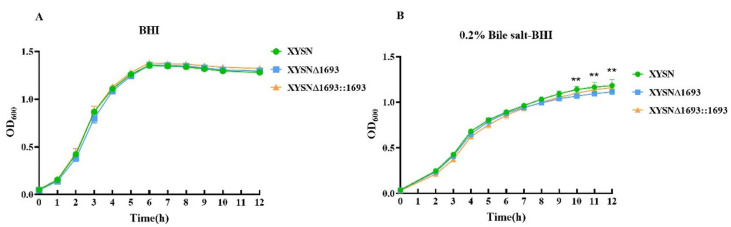
**LMxysn_*1693* gene participates in bile salt resistance of Lm.** The fresh culture of wild type, LMxysn_*1693* mutant and complemented strains were adjusted to OD_600_ = 0.05, they were inoculated to BHI medium and 0.2% bile salt–BHI medium under 37 °C, respectively. The growing curve was measured in BHI medium (**A**). Values are means ± SEM, n = 3 independent experiments. There is no difference among the three strains. The growing curve was measured in 0.2% bile salt-BHI medium (**B**). Values are means ± SEM, n = 3 independent experiments. Statistically significant differences among the deletion strain and parental or reverse mutation strains, and they were determined by Tukey’s multiple comparisons test: ns, no significance; ** *p* < 0.01.

**Figure 3 microorganisms-10-01263-f003:**
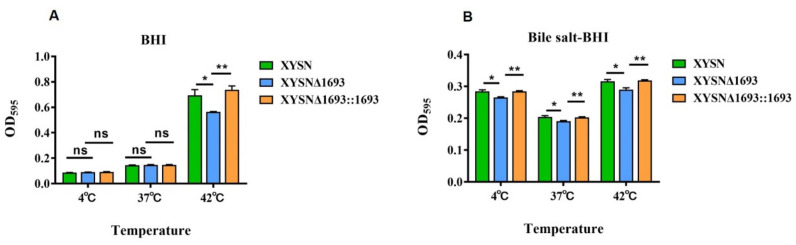
**LMxysn_*1693* gene contributes to biofilm formation.** A quantity of overnight bacterial cultures was added into a sterile 96-well for 72 h at 37 °C. The level (OD) of the crystal violet present in destaining solution was measured at 595 nm in BHI medium at 4 °C, 37 °C and 42 °C (**A**), bile salt-BHI medium at three temperatures (**B**). Each strain was tested in triplicate. They were determined by Student’s *t*-test: ns, no significance; * *p* < 0.05, ** *p* < 0.01.

**Figure 4 microorganisms-10-01263-f004:**
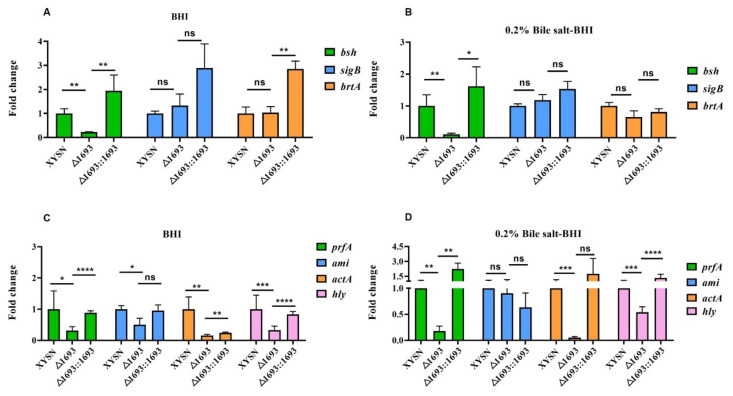
**The expression of bile salt resistance genes and virulence genes by qRT-PCR.** The transcriptional expressions of bile salt resistance genes were determined in BHI medium and 0.2% bile salt-BHI medium (**A**,**B**). The transcriptional expressions of virulence genes were determined in BHI medium and 0.2% bile salt–BHI medium (**C**,**D**). Error bars represented SD, n = 3 independent experiments. Statistical analysis was carried out by Student’s *t*-test: ns, no significance; * *p* < 0.05, ** *p* < 0.01, *** *p* < 0.001, **** *p* < 0.0001.

**Figure 5 microorganisms-10-01263-f005:**
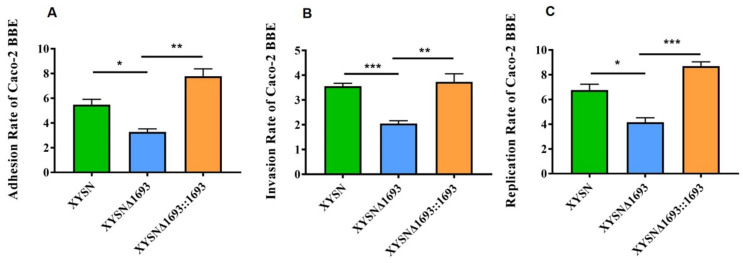
**Adhesion, invasion and proliferation capacities of Lm.** Caco-2 BBe cell was infected (MOI = 20) with Lm XYSN, XYSNΔ*1693* and XYSNΔ*1693*::*1693*, respectively. The percentage of intracellular bacteria was calculated after 15 min of bacteria invasion or two hours of bacteria proliferation. The results are the adhesion (**A**), invasion (**B**) and replication (**C**), respectively. Among them, Lm XYSN is represented by the green bar, the mutant strain is represented by the blue bar and the complemented strain is represented by the orange bar. Error bars represented SD, n = 3 independent experiments. Statistical analysis was carried out by Student’s *t*-test: * *p* < 0.05, ** *p* < 0.01, *** *p* < 0.001.

**Figure 6 microorganisms-10-01263-f006:**
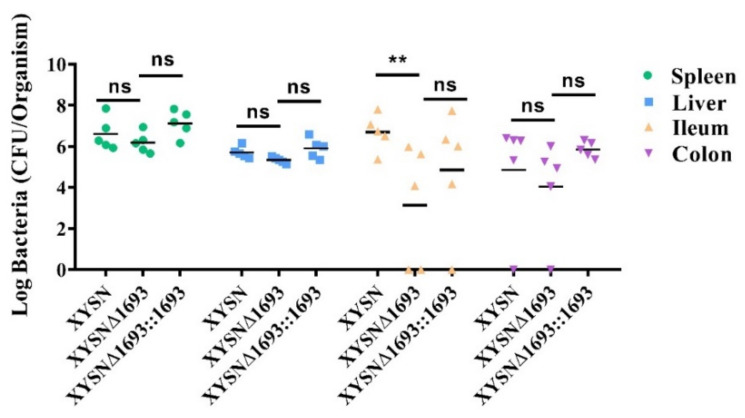
**Bacteria loads in organs after Lm infection mice.** Lm XYSN, XYSNΔ*1693* and XYSNΔ*1693*::*1693* (3 × 10^6^ CFU) were intragastrically inoculated to mice (n = 5). Animals were euthanized 72 h after infection, and organs were recovered, homogenized and plated. The numbers of bacteria able to colonize the ileum, colon, liver and spleen are shown. Among them, spleen is represented by the green circle, liver is represented by the blue square, ileum is represented by the orange triangle and colon is represented by the purple reverse triangle. Values are mean ± SEM; n = 3 independent experiments. Statistical analysis was performed using Tukey’s multiple comparisons test: ns, no significance; ** *p* < 0.01.

**Table 1 microorganisms-10-01263-t001:** Predictions the function of genes in GI-7.

Gene	Domain	Location	Functional Prediction
LMxysn_*1677*	None	Outside	exonuclease SbcD
LMxysn_*1678*	PlsC Domain	Outside	1-acyl-sn-glycerol-3-phosphate acyltransferase
LMxysn_*1679*	Transmembrane Region	TM helix	Lipoprotein
LMxysn_*1680*	Transmembrane Region	TM helix	Hypothetical Protein
LMxysn_*1681*	AAA Domain	TM helix	ABC transporter ATP-binding protein/permease
LMxysn_*1682*	AAA Domain	TM helix	ABC transporter ATP-binding protein/permease
LMxysn_*1683*	Transmembrane Region	TM helix	Cellsurface protein
LMxysn_*1684*	None	Outside	Proteolysis
LMxysn_*1685*	None	Inside	Hypothetical Protein
LMxysn_*1686*	Transmembrane Region	TM helix	Hypothetical Protein
LMxysn_*1687*	None	Inside	Hypothetical Protein
LMxysn_*1688*	Low complexity	Outside	DUF3221 Domain-containing Protein
LMxysn_*1689*	None	Inside	Hypothetical Protein
LMxysn_*1690*	Low complexity	Outside	Hypothetical Protein
LMxysn_*1691*	Transmembrane Region	TM helix	Hypothetical Protein
LMxysn_*1692*	Transmembrane Region	TM helix	Multisubunit Na^+^/H^+^ antiporter
LMxysn_*1693*	Transmembrane Region	TM helix	Hypothetical Protein
LMxysn_*1694*	None	Outside	Hypothetical Protein
LMxysn_*1695*	SH3B Domain	Inside	N-acetylmuramoyl-L-alanine amidase
LMxysn_*1696*	Phage_integrase	Outside	Belongs to the ‘phage’ integrase family

The corresponding bioinformatic predictions of the genes contained in GI-7 are shown in Table 1, including the domains contained in the protein, cellular localization and functional predictions.

## Data Availability

The datasets used and/or analyzed during the current study are available from the corresponding author on reasonable request.

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
