# Peer review of "Transmembrane Protein LMxysn_1693 of Serovar 4h Listeria monocytogenes Is Associated with Bile Salt Resistance and Intestinal Colonization"

_microorganisms, 2022, doi:10.3390/microorganisms10071263_

Round 1
Reviewer 1 Report
In the submitted manuscript, Jin et.al. characterized the function(s) of the genetic element “Lmxysn_1693” in Listeria serotype 4h strain Lm XYSN. The authors showed that this genetic element is in the genomic island-7. Deletion of this gene renders the pathogen less resistant to bile salts and less biofilm formation. Furthermore, the authors demonstrated that this gene is involved in transcriptional regulation of prfA, a master virulence regulator and its regulated genes such as bsh, actA, and hly. Lastly, the authors showed that this mutant has virulence defect in both Caco-2 cell culture model and gastrointestinal tract (ileum) of the infected mice. Overall, the study is well conducted and will provide useful information to the audiences in the field.
Here are some minor points for the authors to consider:
Line 204: change “consistent with” to “similar when”
Line 204-208: overall, the defeat of mutant in bile salt condition is mild. I would suggest the authors to tone down the conclusion and remove the word “strikingly”.
Line 227-228: it is hard to prove the causality between biofilm formation and the bile salt resistance, could be vise versa. I would suggest the authors to conclude that mutant forms less biofilm in bile salt conditions.
Line 341: change “paly” to “play”.
Author Response
回复审稿人 1 条评论
第 204 行:将“consistent with”更改为“similar when”
我们同意您的意见,并对稿件内容进行了修改,修改如下。
209-210行:XYSN、 XYSNΔ1693和XYSNΔ1693 :: 1693的生长曲线(图2A)在BHI中37℃培养时除两个时间点外基本相似。
第 204-208 行:总体而言,突变体在胆盐条件下的失败是温和的。我建议作者淡化结论并删除“惊人地”一词。
上述句子修改如下。
Line 210-214: While the growth of mutant was significantly inhibited in the stationary phase when it was cultured in the BHI supplemented with 0.2% bile salt (Fig. 2B). The treat of bile salts affected the growth of mutant in the stationary phase, which indicated that LMxysn_1693 contributed to resist stress response.
Line 227-228: it is hard to prove the causality between biofilm formation and the bile salt resistance, could be vise versa. I would suggest the authors to conclude that mutant forms less biofilm in bile salt conditions.
We have revised the sentence.
Line 233-234: These results indicated that deletion of LMxysn_1693 formed less biofilm in bile salt conditions, therefore decreasing resistance to bile salts stress.
Line 341: change “paly” to “play”.
“paly”一词改为“play”。
第 347 行:ABC 转运蛋白在细菌金属利用、生物膜形成、抗各种压力、毒力和感染等方面发挥多种功能。

Reviewer 2 Report
The manuscript of F. Jin et al. titled "Transmembrane Protein LMxysn_1693 of 4h Listeria monocytogenes is Associated with Bile Salt Resistance and Intestinal Colonization" is devoted to the characterization of novel transmembrane protein of genomic island 7 of L. monocytogenes serovar 4h and its role in intestinal colonization and biofilm formation. The results are well supported by experimental work. The authors are convinced of the role of the protein in membrane transport.
The major issue of the manuscript is the English usage because English is not the native language of the authors. A thorough English editing is needed. Some sentences are not understandable.
Other minor issues are given below:
Line 2: I suggest rephrasing it as "Listeria monocytogenes serovar 4h" since 4h before the microorganism name is not clear, I can understand it as L. monocytogenes after 4 hours of growing on media... The same for line16: put serovar after the Latin name "Listeria monocytogenes serovar 4h".
Line 7-12: Omit brackets.
Line 13: Do not bold Listeria
Line 33: You wrote about hypervirulent clonal complexes, but what about hypervirulent serovars, such as 4h and others? How many CCs does it have? Only CC33? Add this here.
Line 34: Typo: lineage
Line 59: Omit And at the beginning of the sentence.
Line 60: 4b is not a clonal complex, but serovar.
Line 62: Serovars of Salmonella must be capitalized and not italicized.
Line 64-66: Add reference for the sentence.
Line 68-70: Add reference for the sentence.
Line 74: I can't find the Statistical Analysis section in M&M. Add it.
Line 78: Typo: Dickinson
Line 95: Replace 1693 with LMxysn_1693
Line 114: Replace this line with 2 GN ID card (Biomerieux, France).
Line 118: Typo: Germany
Line 119: What volume of media is in flasks?
Line 120: How long it was measured: for 10h, 12h? It is different in figures 2A and 2B.
Line 130: Which enzyme-labeled instrument was used?
Line 139: Capitalize b in gyrB
Line 143: Typo: stabilizer
Line 161: How it was improved? Maybe you meant modified Oxford medium?
Lines 213, 215: What is show, there has?
Line 282: Typo: CFUs
Line 298-301: Rephrase this sentence. I can not understand what you meant?
Line 302-304: Do not ilalisize PrfA, SigB, LuxS.
Line 324: Move pump before MdrT
Line 324-325: Rephrase this sentence.
Line 341: Typo: play
Add a list of all supplement figures and tables!
Author Response
Response to Reviewer 2 Comments
Dear reviewer:
We are truly grateful for your valuable comments on our work. The manuscript has been revised according your feedback, a point-by-point response to your comments and an overview of the modifications introduced in the manuscript can be found below.
Line 2: I suggest rephrasing it as "Listeria monocytogenes serovar 4h" since 4h before the microorganism name is not clear, I can understand it as L. monocytogenes after 4 hours of growing on media... The same for line16: put serovar after the Latin name "Listeria monocytogenes serovar 4h".
Putting serovar 4h before Listeria monocytogenes is the normal writing format, serovar 4h is used to modify the noun of “Listeria monocytogenes”.
For example: Sumrall ET, Schefer CRE, Rismondo J, Schneider SR, Boulos S, Gründling A, Loessner MJ, Shen Y. Galactosylated wall teichoic acid, but not lipoteichoic acid, retains InlB on the surface of serovar 4b Listeria monocytogenes. Mol Microbiol. 2020, 113, 638-649.
Ueda F, Yugami K, Mochizuki M, Yamada F, Ogasawara K, Fujima A, Shoji H, Hondo R. Comparison of genomic structures in the serovar 1/2a Listeria monocytogenes isolated from meats and listeriosis patients in Japan. Jpn J Infect Dis. 2005, 58, 289-293.
Buncic S, Avery SM, Rocourt J, Dimitrijevic M. Can food-related environmental factors induce different behaviour in two key serovars, 4b and 1/2a, of Listeria monocytogenes? Int J Food Microbiol. 2001, 65, 201-12.
Doijad S, Lomonaco S, Poharkar K, Garg S, Knabel S, Barbuddhe S, Jayarao B. Multi-virulence-locus sequence typing of 4b Listeria monocytogenes isolates obtained from different sources in India over a 10-year period. Foodborne Pathog Dis. 2014, 11, 511-516.
Line 7-12: Omit brackets.
The brackets are removed now.
Line 7-12: Jiangsu Key Laboratory of Zoonosis, Key Laboratory of Prevention and Control of Biological Hazard Factors (Animal Origin) for Agrifood Safety and Quality, MOA of China, Joint International Research Laboratory of Agriculture and Agri-Product Safety, Jiangsu Co-Innovation Center for Prevention and Control of Important Animal Infectious Disease and Zoonosis, Yangzhou University, 48 East Wenhui Road, Yangzhou 225009, Jiangsu Province, China
* Correspondence: yylan@yzu.edu.cn
Line 13: Do not bold Listeria
The sentence has been corrected.
Line 13-14: Listeria monocytogenes (Lm) is a ubiquitous foodborne pathogen comprising 14 serotypes, of which serovar 4h isolates belonging to hybrid sub-lineage Ⅱ exhibit hypervirulent features.
Line 33: You wrote about hypervirulent clonal complexes, but what about hypervirulent serovars, such as 4h and others? How many CCs does it have? Only CC33? Add this here.
All serovar 4h isolates are belonged to CC33, except 4h strains, no other strains are belonged to CC33.
Line 34: Typo: lineage
The typo mistake is corrected.
Line 34-37: CC33, consists of hybrid sub-lineage Ⅱ (HSL-Ⅱ) hypervirulent isolates, is an emerging complex clone with 200-400 folds higher virulence than Lm EGD-e via oral infection which is highly susceptible to wild type mice.
Line 59: Omit And at the beginning of the sentence.
We agree with your comment and have revised the sentence.
Line 59-61: Lee et al. identified that a 4b clonal complex could resist arsenic and cadmium with a 35 kb genomic island.
Line 62: Serovars of Salmonella must be capitalized and not italicized.
The writing format of Serovars of Salmonella is capitalized, we have amended this writing format.
Line 62-64: For example, Salmonella Senftenberg could survive under glucose as the sole carbon source, mainly due to the metabolic island CTnscr94 with related genes, which is crucial for enhancing environmental adaptability.
Line 64-66: Add reference for the sentence.
The reference is added after the sentence.
Line 64-67: Undoubtedly, CC33 isolate Lm XYSN harbor 8 genomic islands essentially involved in the adaption of saprophytic and parasitic life. Understanding functions of genes in the genomic islands contribute to deciphering the pathogenic mechanism from virulence to hypervirulence [6].
Line 68-70: Add reference for the sentence.
The sentence of Line 68-70 is previously unreported data, the conclusion was obtained from this work, therefore no relevant references can be added to support it.
Line 74: I can't find the Statistical Analysis section in M&M. Add it.
We have added the statistical analysis section in the methods.
Line 163-168:
2.10. Date Analysis
The results are expressed as the means ± SD for the results of all qRT-PCR. The results of detecting adhesion, invasion and replication capacity of Lm are as well. Values are means ± SEM for the results of growth curves and in vivo experiments. Data were analyzed by Student’s t-test or Tukey’s multiple comparisons test. Differences were considered significant at ns, none significance, * p ≤ 0.05, ** p ≤ 0.01, *** p ≤ 0.001 and ****p < 0.0001.
Line 78: Typo: Dickinson
The word is corrected.
Line 78: All Lm strains were cultured in brain heart infusion (BHI; Becton Dickinson, USA).
Line 95: Replace 1693 with LMxysn_1693
1693 is replaced by LMxysn_1693 in this sentence.
Line 95-97: Amino acid sequence of LMxysn_1693 protein was submitted to the SWISS-MODEL to obtain the hypothetical 3D structure (http://swissmodel.expasy.org).
Line 114: Replace this line with 2 GN ID card (Biomerieux, France).
2 GN ID card is rewritten by VITEK® 2 GN ID card.
Line 112-114: The bacterial turbidity was controlled at ∼0.5 with a nephelometer and biochemical characteristics were identified using a VITEK® 2 GN ID card (Biomerieux, France).
Line 118: Typo: Germany
The word German is replaced by Germany.
Line 116-118: Bacterial of exponentially growing cultures of Lm XYSN, XYSNΔ1693 and XYSNΔ1693::1693 were harvested and adjusted to an initial OD600 value of 0.05 in 20 mL BHI culture and 20 mL 0.2% bile salt (Merck Sigma, Germany) in BHI culture respectively.
Line 119: What volume of media is in flasks?
The detailed volume of 20 mL media in the flasks is updated.
Line 118-119: Three parallel replicates were set for each strain and each treatment with 20 mL media.
Line 120: How long it was measured: for 10h, 12h? It is different in figures 2A and 2B.
The total time for measurement of growth curve was 12h. This sentence is amended as follows.
Line 118-119: Three parallel replicates were set for each strain and each treatment with 20 mL medium for 12h.
Line 130: Which enzyme-labeled instrument was used?
The type of this enzyme-labeled instrument is added.
Line 130-131: The biofilm formation was measured at 595 nm by BioTek synergy 2 enzyme-labeled instrument (BioTek, America).
Line 139: Capitalize b in gyrB
The word gyrb is changed to gyrB.
Line 139-140: The house-keeping gene gyrB was selected as the internal reference gene to reflect the differences in gene expression objectively.
Line 143: Typo: stabilizer
The wrong word is amended.
Line 143-144: Human intestinal epithelial cell line Caco-2 BBe with stabilized growth conditions was seeded into 24-well cell culture plates.
Line 161: How it was improved? Maybe you meant modified Oxford medium?
The description is not accurate. We have added the detailed information of the Oxford medium.
Line 159-162: The organs were homogenized and plated serial dilutions on the BHI plate, the intestine samples were spread onto modified Oxford medium (BD Diagnostics, USA) chromogenic plate, then the bacterial loads were counted after 20 h post spreading.
Lines 213, 215: What is show, there has?
The sentence is rewritten as follows.
Line 219-220: Values are means ± SEM, n = 3 independent experiments. There is no difference among the three strains.
Line 282: Typo: CFUs
The misspelled word is corrected.
Line 287-289: After 72 hours post-infection, the spleen, liver, colon and ileum of the mice were homogenized and CFUs was counted, for which the results were analyzed through the Tukey’s multiple comparisons test.
Line 298-301: Rephrase this sentence. I can not understand what you meant?
The sentence is revised.
Line 304-307: Lm is a foodborne pathogen that has evolved variety strategies to survive in the gastro-intestinal environment, secretion extracellular polysaccharides to form biofilm is one of the ways to resist the stress factors in the gut environment.
Line 302-304: Do not ilalisize PrfA, SigB, LuxS.
The writing format of ilalisize PrfA, SigB, LuxS is corrected.
Line 307-310: Previous studies have demonstrated that transcription activators PrfA, SigB, DegU, quorum sensing (QS) system, and accessory gene regulatory system (Agr) operon could up-regulate the biofilms formation, while it is down-regulated by ABC transfusion permease and LuxS.
Line 324: Move pump before MdrT
The words pump is moved before MdrT.
Line 328-330: Bile sensor BrtA controls expression of the cholic acid efflux pump MdrT, and CadC represses BSH expression to avoid the over expression of the cholic acid efflux pump MdrT.
Line 324-325: Rephrase this sentence.
We have rephrased this sentence.
Line 330-331: Whereas, the mechanism to resistant bile or bile salts for Lm is not fully understood.
Line 341: Typo: play
The typo mistake is corrected.
Line 347: ABC transporters play multiple functions in bacterial metal utilization, biofilm formation, resistance to various stress, virulence and infection.
Add a list of all supplement figures and tables!
In fact, we have uploaded all supplementary materials to the system in the form of compressed packages, but we have not obtained the internet site.
Now we have added the description of all supplement figures and tables (Line 377 to Line 385), additionally, all supplement figures and tables are listed from Line 503 to 534.
